# $Ag_9GaSe_6$: high-pressure-induced Ag migration causes thermoelectric performance irreproducibility and elimination of such instability

Jing-Yuan Liu [1], Ling Chen [1,2✉] & Li-Ming Wu [1,2✉]

The argyrodite $Ag_9GaSe_6$ is a newly recognized high-efficiency thermoelectric material with an ultralow thermal conductivity; however, liquid-like Ag atoms are believed to cause poor stability and performance irreproducibility, which was evidenced even after the 1st measurement run. Herein, we demonstrate the abovementioned instability and irreproducibility are caused by standard thermoelectric sample hot-pressing procedure, during which high pressure promotes the 3-fold-coordinated Ag atoms migrate to 4-fold-coordinated sites with higher-chemical potentials. Such instability can be eliminated by a simple annealing treatment, driving the metastable Ag atoms back to the original sites with lower-chemical potentials as revealed by the valence band X-ray photoelectron chemical potential spectra and single crystal X-ray diffraction data. Furthermore, the hot-pressed-annealed samples exhibit great stability and TE property repeatability. Such a stability and repeatability has never been reported before. This discovery will give liquid-like materials great application potential.

[1] Beijing Key Laboratory of Energy Conversion and Storage Materials, College of Chemistry, Beijing Normal University, Beijing 100875, People's Republic of China. [2] Center for Advanced Materials Research, Advanced Institute of Natural Sciences, Beijing Normal University, Zhuhai 519087, People's Republic of China. ✉email: chenl@bnu.edu.cn; wlm@bnu.edu.cn

Thermoelectric (TE) materials, which can not only realize the mutual transformation of heat and electric energy but also exhibit the advantages of no emission, no noise and long lifetime, have attracted intense and continuous attention[1–6]. The energy conversion efficiency of a TE material is defined by the figure of merit ($ZT$), $ZT = S^2\sigma T/\kappa$, where $S$, $\sigma$, $T$ and $\kappa$ represent the Seebeck coefficient, electrical conductivity, absolute temperature and thermal conductivity ($\kappa$ consists of electronic thermal conductivity $\kappa_e$ and lattice thermal conductivity $\kappa_l$), respectively. Among these parameters, $\kappa_l$ is relatively independent and easy to regulate; thus, an effective strategy for designing and searching new TE materials with high performance is to find materials with low intrinsic $\kappa_l$[7–11]. Argyrodite family with a general formula of $A_{12-n}^{+}B^{n+}X_6^{2-}$, ($A^+ = Li^+$, $Cu^+$ or $Ag^+$, $B^{n+} = Al^{3+}$, $Ga^{3+}$, $Si^{4+}$, $Ge^{4+}$, $Sn^{4+}$, $P^{5+}$, $As^{5+}$, $Nb^{5+}$ or $Ta^{5+}$ and $X^{2-} = S^{2-}$, $Se^{2-}$, and $Te^{2-}$) has been recently recognized as a high TE performance material showing intrinsic low $\kappa_l$[12]. Argyrodite crystallizes in a cubic or hexagonal structure with a disordered A-cation sublattice[13]. Taking the representative $Ag_9GaSe_6$ as an example, the typical structural feature is an anion framework of $[GaSe_6]^{9-}$ providing disordered Ag cation sites, within which the Ag atoms may be mobile from site to site. Such migration is known as a liquid-like behavior, by the definition of the concept of "Phonon-Liquid Electron-Crystal (PLEC)"[2]. In particular, such liquid-like behavior in a TE material not only strongly scatters the phonons but also softens some of the transverse phonons, resulting in an ultralow $\kappa_l$ and high $ZT$[2]. Thus, argyrodites feature ultra-low $\kappa_l$ values (ranging from 0.15 to 0.45 W m$^{-1}$ K$^{-1}$) and high $ZT$ values (approximately 1.1)[12]. For example, these values were found in many of its members, $Ag_9GaSe_6$ (Fig. 1), $Ag_9GaS_6$, $Ag_9GaTe_6$, $Ag_9AlSe_6$, $Ag_8GeSe_6$, $Ag_8SnSe_6$, $Ag_8GeTe_6$, $Ag_8SiTe_6$, $Cu_8GeSe_6$, and $Cu_7PSe_6$, among which the $Ag_9GaSe_6$-based material is the best n-type liquid-like TE material known to date[5,12,14–28].

Normally, the p-type liquid-like TE material is superior to its n-type counterpart in performance and is rich in the number of available candidates. As listed in Supplementary Table 14, the typical p-type liquid-like TE materials (such as the $Cu_2Q$-based compounds[5], Cu-argyrodite[21–25], and superionic conductors[29–36])

are more abundant than the n-type counterparts (e.g., $Ag_2Q$-based compounds[37,38], Ag-argyrodite[14–20,26–28].) Besides, the best p-type liquid-like TE material can realizes a $ZT$ of 2.6 in $Cu_2Se/CuInSe_2$;[39] whereas the best n-type one can only reaches a $ZT$ of 1.6 by $Ag_9GaSe_6$[16,26,27]. Thus, to explore on the n-type liquid-like materials are of great significance.

Pristine $Ag_9GaSe_6$ exhibits a low $\kappa_l = 0.15$ W m$^{-1}$ K$^{-1}$ and a high $ZT_{max} = 1.1$ at 800 K, almost the lowest $\kappa_l$ and the highest $ZT$ values among all n-type liquid-like materials known to date[15,16]. Since the first report on $Ag_9GaSe_6$ in 2017, a flow of subsequent efforts flooded to further optimize the TE performance; for example, a slight composition-off-stoichiometry Se deficiency ($Ag_9GaSe_{5.98}$) effectively increased the carrier concentration to $5.1 \times 10^{18}$ cm$^{-3}$ at 300 K by providing more electrons, which is approximately one order of magnitude higher than that of pristine $Ag_9GaSe_6$, leading to a $ZT_{max}$ of 1.3 at 800 K;[15] additionally, alloying Te at the Se site reduced the carrier concentration to $3 \times 10^{18}$ cm$^{-3}$, thus reducing $\kappa_e$, resulting in an increased $ZT_{max} = 1.5$ at 850 K[16]. In addition, alloying Te at the Se sites increases the configurational entropy, which reduces $\kappa_l$ by introducing extra phonon disorder due to the presence of strong extra mass and strain fluctuations, giving $Ag_9GaSe_{5.98}$ a $ZT_{max} = 1.6$ at 850 K[26]. Furthermore, alloying Cu at the Ag site decreases the carrier concentration and eventually allows $Ag_{8.28}Cu_{0.72}GaSe_6$ to have a $ZT_{max}$ of 1.6 at 824 K[27].

However, similar to other liquid-like materials, $Ag_9GaSe_6$ also shows typical, characteristic, yet troublesome features, i.e., instability and TE performance irreproducibility upon an external electric field or a temperature gradient. For example, $Cu_2Q$ tends to deposit Cu metal on the pellet surface during property measurement, which leads to severe performance degradation[40–45]. Of note, the majority of studies on argyrodites merely report the 1st-run test results of the TE properties[14–26]. Luo et al. even pointed out that the TE properties of $Ag_9GaSe_6$ are extremely unrepeatable, i.e., from the 1st run to the 2nd run of the measurement, $\sigma$ increases 6-fold from 100 to 700 S cm$^{-1}$ (Fig. 2a); simultaneously, $\kappa$ increases from 0.35 to 0.75 W m$^{-1}$ K$^{-1}$ (Fig. 2c), leading to a dramatic $ZT$ reduction (1.1 to 0.5 at 700 K,

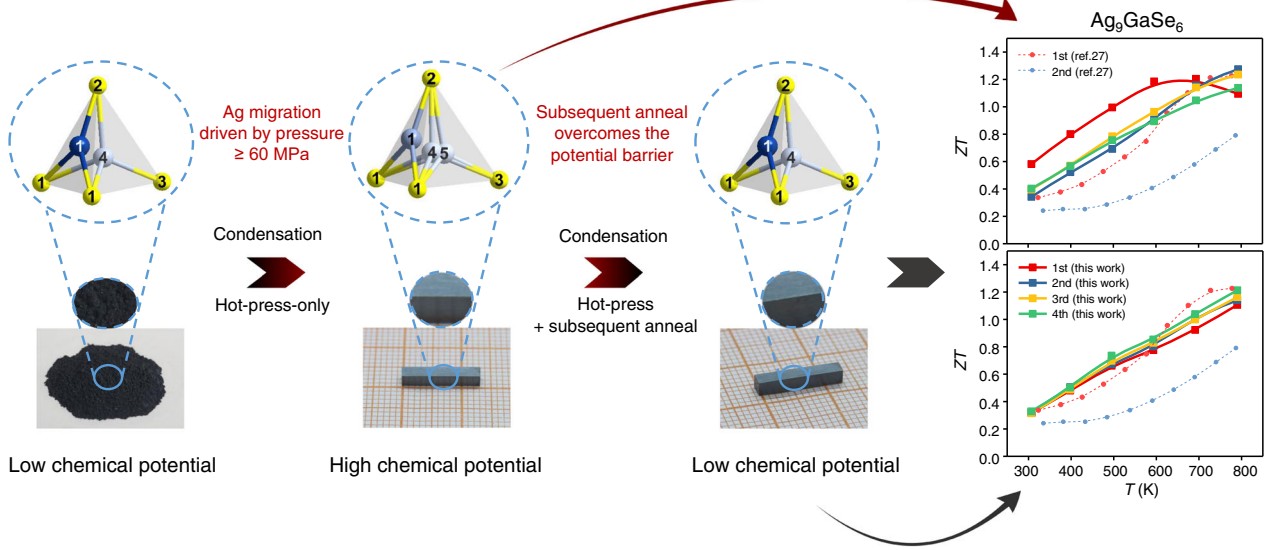

**Fig. 1 Schematic illustration of the hot-pressed-annealed condensation process of the $Ag_9GaSe_6$-based n-type thermoelectric materials that exhibit unprecedented stability and reproducibility, which suggests that they have a great application potential as excellent n-type TE materials.** Standard hot-pressing promotes Ag atoms to migrate to the tetrahedron center, a high chemical potential site with larger coordination number. Ag atoms located at these metastable sites tend to be easily reduced, leading to the well-known TE property irreproducibility. We discovered that a simple hot-pressed-annealed treatment helps metastable Ag atoms to overcome the potential barrier and migrate back to the original low-chemical-potential sites, and consequently show an unprecedented TE property repeatability.

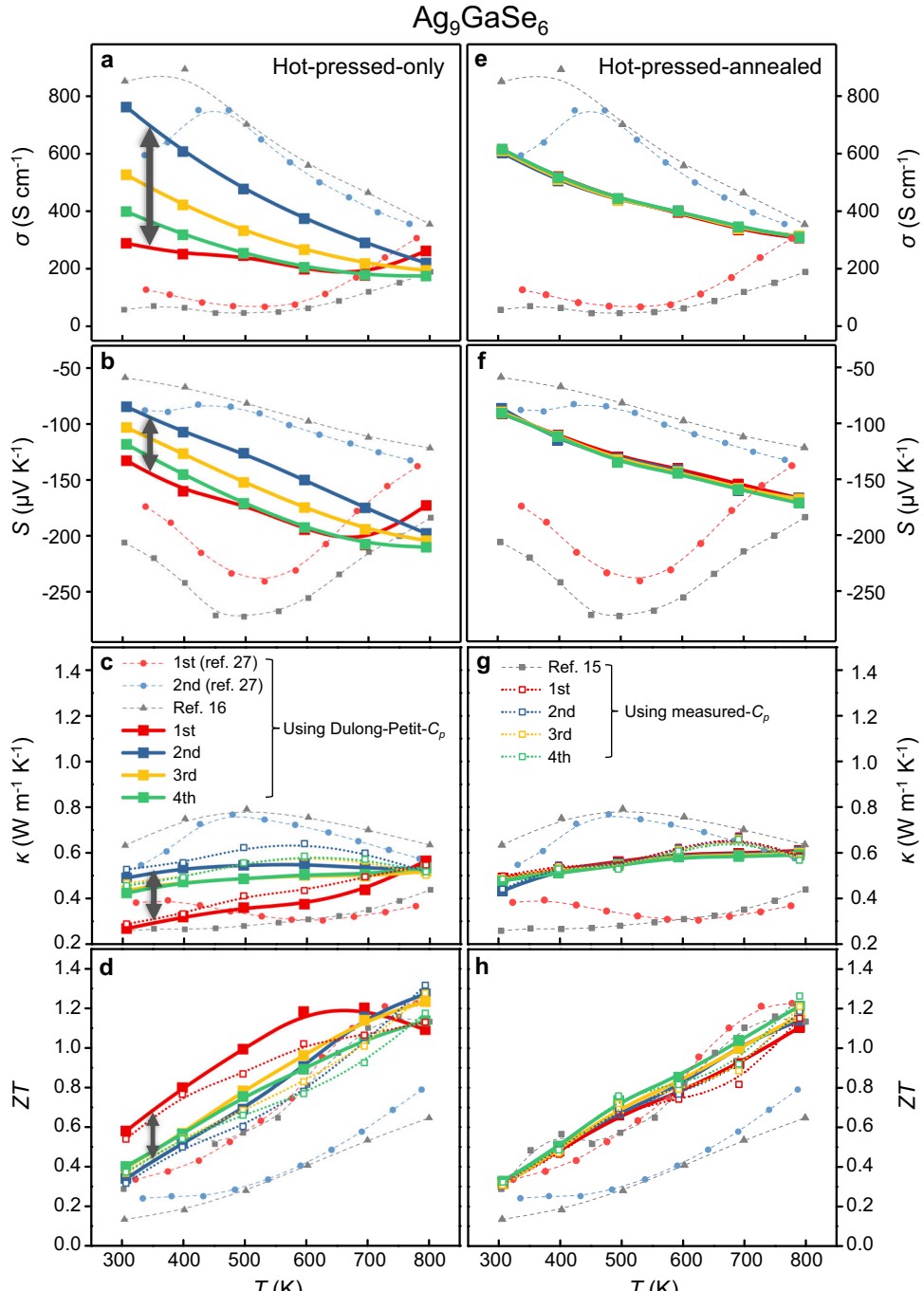

**Fig. 2 Temperature-dependent thermoelectric properties of Ag$_9$GaSe$_6$. a–d** hot-pressed-only, and **e–h** hot-pressed-annealed samples. **a, e** Electrical conductivity $\sigma$. **b, f** Seebeck coefficient $S$. **c, g** Thermal conductivity $\kappa$. **d, h** Figure of merit $ZT$. The $\kappa$ and $ZT$ calculated by both the measured-$C_p$ and Dulong-Petit-$C_p$ and those reported[15,16,27] are presented. The gray double arrows indicate the significant discrepancy observed on the hot-pressed-only samples. The measured-$C_p$ are provided in Supplementary Information Supplementary Fig. 2.

Fig. 2d)[27]. This irreproducibility was ascribed to the enrichment of Ag metal in the hot side of the pellet caused by Ag migration[27]. Similarly, Ag$_8$SnSe$_6$ shows dramatically different electrical properties, a semiconducting $S = -500$ μV/K in the heating run but a metallic $S = -100$ μV/K in the cooling run[28]. Such TE performance irreproducibility was constantly reported and was believed to be attributed to the migration of liquid-like Ag atoms upon external stimulation. Even worse, such unrepeatability makes argyrodite, or even the class of "liquid-like" TE materials, face very slim and vague application prospects.

Herein, we demonstrate that the abovementioned instability and irreproducibility are caused by standard thermoelectric sample processing, i.e., the hot-press procedure, during which high pressure (>60 MPa) induces a metastable Ag distribution, which is proven to have a high chemical potential and leads to performance instability (Fig. 1). We show that such instability can be eliminated by a simple annealing treatment on the hot-pressed pellet at 823 K for 24 h that drives the metastable Ag atoms back to the original room temperature crystallographic sites, as revealed by the single crystal X-ray diffraction data. These sites

are energetically more stable low-chemical potential sites, as suggested by the valence band X-ray photoelectron chemical potential spectra. Furthermore, we present annealed samples showing great stability and TE property repeatability. More remarkably, the hot-pressed-annealed Cu/Te-doped $Ag_9GaSe_6$ sample actually maintains a $ZT = \sim 1.4$, one of the highest values for this family to date, even after six runs of measurements, which has never been reported before. This discovery will give argyrodite and other related liquid-like materials great application potential.

## Results

Pure-phased $Ag_9GaSe_6$-based samples were synthesized by solid-state reactions accordingly[16,17,19,20,25], which were condensed via two different procedures, as illustrated in Fig. 1. First, as a representative, the as-synthesized pure phased $Ag_9GaSe_6$ powder (Supplementary Fig. 5) was condensed by standard thermoelectric sample processing, i.e., hot pressing, and the obtained sample is denoted as a hot-pressed-only sample hereafter. Subsequently, the TE properties were measured (Fig. 2a–d). In agreement with the literature, both the electrical conductivity ($\sigma$), Seebeck coefficient ($S$), thermal conductivity ($\kappa$) and $ZT$ of the hot-pressed-only $Ag_9GaSe_6$ sample exhibit poor stability and repeatability. For example, at $T = 400$ K, the relative standard deviation (RSD) of $\sigma$ is a typical 34% between the 1st and 2nd runs of the measurement, which is much higher than the measurement uncertainty ($\sim 5\%$). The RSDs of $S$, $\kappa$ and $ZT$ are 15%, 17 and 18%, respectively. Such discrepancies have been repeatedly observed previously, and they were attributed to Ag migration caused by external stimulation during the measurement[27,28]. Besides, the scanning electron microscopy observation reveals that the Ag-rich precipitation is observed on the hot-pressed-only-$Ag_9GaSe_6$ after the TE-property measurement (Supplementary Figs. 7a vs b), manifesting the sample instability as similar as that observed by Luo[27]. Remarkably, herein, we found for the first time that such instability and irreproducibility can be eliminated by annealing the hot-pressed-only sample at 823 K for 24 h under $N_2$ flow. The as-treated sample is denoted as the hot-pressed-annealed sample hereafter. As shown in Fig. 2e–h, in comparison with the hot-pressed-only sample as well as those reported in the literature, the hot-pressed-annealed sample exhibits unprecedented stability and TE performance reproducibility, and the RSDs of $\sigma$, $S$, $\kappa$ and $ZT$ are approximately 2% within the uncertainty of the instruments. Figure 2e–h suggest typical degenerate semiconducting and n-type TE material behavior of hot-pressed-annealed $Ag_9GaSe_6$, which is consistent with those previously reported[15,16,26,27]. $\sigma$ ranges from 600 to 300 S cm$^{-1}$ (Fig. 2e); negative $S$ monotonically varies with increasing temperature (Fig. 2f), and $\kappa$ changes slightly in the range of 0.5 to 0.6 W m$^{-1}$ K$^{-1}$ as a function of the temperature (Fig. 2g), which are all agreeable in the reasonable range with previous reports[15,16,26,27]. In contrast, the hot-pressed-annealed $Ag_9GaSe_6$ sample unprecedentedly exhibits a constant $ZT_{max} = 1.15$ (Fig. 2h), which is one of the highest values ever reported[30]. More remarkably, such striking performance repeatability has never been observed. Moreover, the SEM observation reveals no obvious Ag precipitation in the hot-pressed-annealed sample before and after the TE property measurements, indicating the phase and composition stability of the samples. (Supplementary Fig. 7c vs d)

According to the single parabolic band (SPB) model, $Ag_9GaSe_6$ is predicted to have a $ZT_{max} = 1.6$ at 800 K if the carrier concentration is adjusted to the $10^{18}$ cm$^{-3}$ level[16], which was confirmed by subsequent experiments by doping Te at the Se site[16,26] or doping Cu at the Ag site[27]. However, the material instability

and property irreproducibility remain severe[27]. To further investigate the validity of the hot-pressed-annealed process, two representatives, the optimized $Ag_9GaSe_{5.5}Te_{0.5}$ and $Ag_{8.3}Cu_{0.7}GaSe_6$ samples with better TE performance[16,27], were selected. These doped samples were successfully synthesized and condensed as described above. As expected, the hot-pressed-only samples still show severe irreproducibility of the TE properties that were in great agreement with the previous report; for example, the RSDs for $\sigma$, $S$, $\kappa$, and $ZT$ at 400 K for $Ag_9GaSe_{5.5}Te_{0.5}$ are 90%, 45%, 23%, and 36%, respectively, (Fig. 3a–d) and for $Ag_{8.3}Cu_{0.7}GaSe_6$, 53%, 25%, 6%, and 14%, respectively. (Fig. 4a–d). In contrast, those treated by the hot-pressed-annealed process exhibit amazing TE property reproducibility; even after the measurement for six runs, great stability was shown, which has never been previously reported. The hot-pressed-annealed $Ag_9GaSe_{5.5}Te_{0.5}$ sample exhibits very low RSDs, which are less than 5% for $\sigma$, $S$, $\kappa$, and $ZT$ at 400 K and constantly achieves $ZT_{max} = 1.3$ at 800 K, which is the best value ever reported[16]. (Fig. 3e–h) Similar phenomenon is observed for the Cu-doped $Ag_{8.3}Cu_{0.7}GaSe_6$ hot-pressed-annealed sample with RSDs of $\sigma$, $S$, $\kappa$, and $ZT$ of 7%, 5%, 5%, and 6% at 400 K, respectively. The hot-pressed-annealed $Ag_{8.3}Cu_{0.7}GaSe_6$ constantly realizes a $ZT_{max} = 1.4$, which is the best value ever reported[27]. (Fig. 4e–h) Moreover, the critical voltage ($V_c$) was measured to be 0.05 and 0.07 V on the hot-pressed-annealed-$Ag_9GaSe_6$ and the hot-pressed-annealed-$Ag_9GaSe_{5.5}Te_{0.5}$, respectively, (Supplementary Fig. 10) which lies between those of the state-of-the-art liquid-like TE materials, $Cu_2S$ (0.02 V) and $Cu_2Se$ (0.11 V)[42], indicating that the subsequent annealing treatment enhances the sample stability. (Supplementary Fig. 11) These solid data and sharp comparison demonstrate that the long-believed yet troublesome TE property instability and irreproducibility of the $Ag_9GaSe_6$-based materials are not intrinsic to the nature of the compound and can be eliminated by a simple process by annealing the hot-pressed pellet at 823 K for 24 h under $N_2$ flow, which can dramatically improve the TE performance repeatability of both the pristine and doped samples. More importantly, such performance reproducibility has never been seen in any liquid-like material before; thus, reconsideration regarding the application potential for such high-TE-performance liquid-like materials is now encouraged.

Furthermore, we try to probe the intrinsic reason and the mechanism of why such annealing-promoted stabilization occurs. The powder XRD patterns of the as-synthesized, hot-pressed-only and hot-pressed-annealed $Ag_9GaSe_6$ samples are all well-indexed, and all peaks are assigned to the cubic phase (PDF#71-2448, space group $F\bar{4}3m$) without any detectable impurities or Ag metal particles, which suggests that the different condensation processes herein do not cause decomposition (Supplementary Fig. 5a). Moreover, with the aid of the internal Si standard, the observed unit cell parameters merely change by a factor less than 0.09‰ in the $a$ parameter (11.1448, 11.1436 vs. 11.1446 Å) and 0.3‰ in $V$ (1384.24, 1383.80 vs. 1384.17 Å$^3$) (Supplementary Fig. 5b). Although the phase identity is the same, the TE performance of these samples is dramatically different. Subsequently, we explore the reason at the atomic level.

Fortunately, we have successfully grown single crystals with excellent crystalline quality in all batches of $Ag_9GaSe_6$ samples. To obtain a statistical view, we manually picked 9 crystals from each batch of samples; on every one of them, single crystal diffraction data were collected: as-synthesized (Supplementary Tables 1, 4 and 7), hot-pressed-only (Supplementary Tables 2, 5 and 8), and hot-pressed-annealed (Supplementary Tables 3, 6 and 9). These single crystal diffraction data reveal comprehensive and reasonable Ag migration upon external stimuli during the condensation treatment. As expected, these crystals are all crystallized in the $F\bar{4}3m$ space group and successfully refined as $Ag_9GaSe_6$

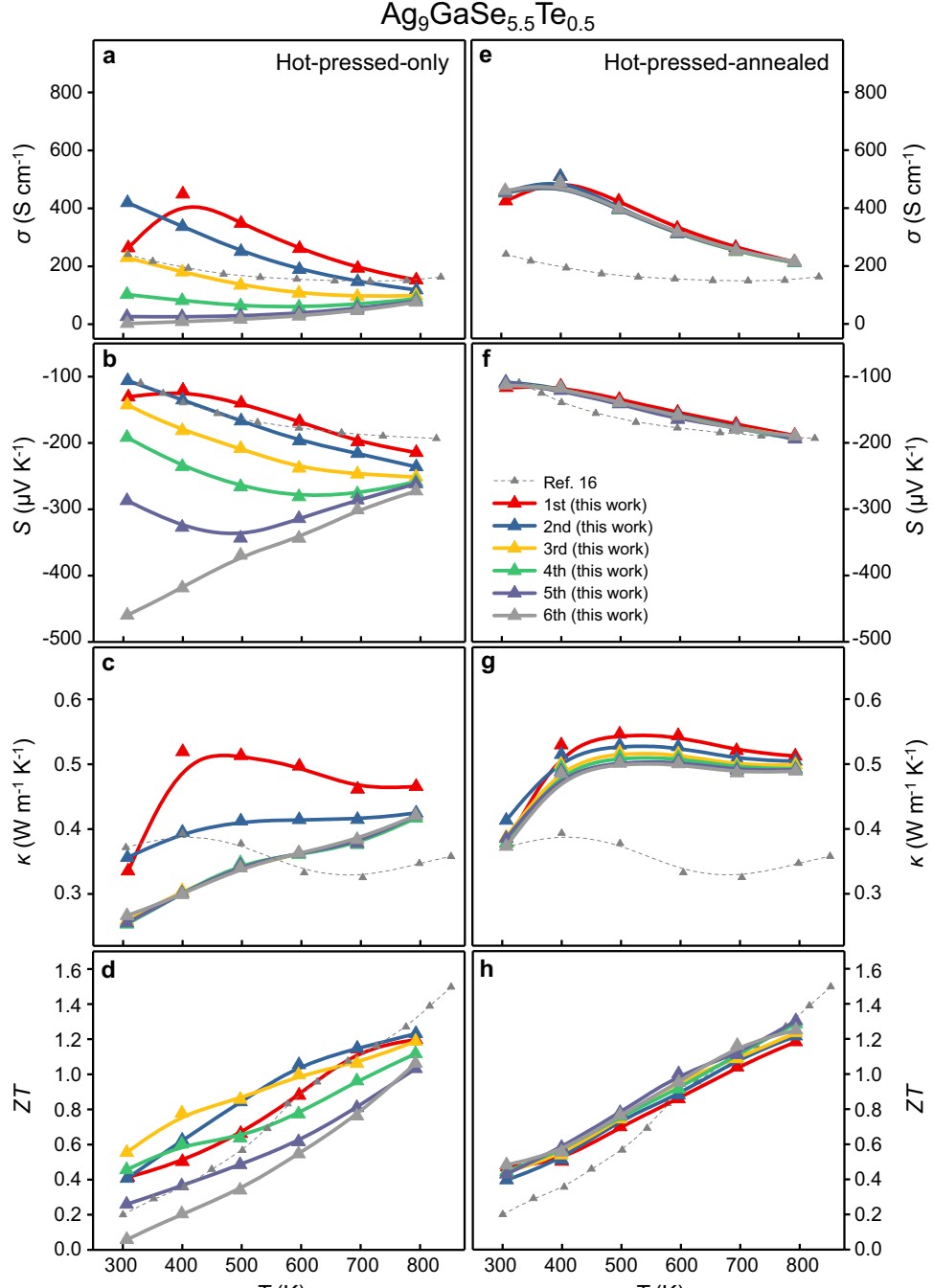

**Fig. 3 Temperature-dependent thermoelectric properties of doped Ag$_9$GaSe$_{5.5}$Te$_{0.5}$. a–d** hot-pressed-only, and **e–h** hot-pressed-annealed samples. **a, e** Electrical conductivity $\sigma$. **b, f** Seebeck coefficient $S$. **c, g** Thermal conductivity $\kappa$. **d, h** Figure of merit $ZT$. The Dulong-Petit-$C_p$ was used. Those reported in ref. [16] that are calculated by the Dulong-Petit-$C_p$ are presented as references. The $\kappa$ and $ZT$ calculated by the measured-$C_p$ are provided in Supplementary Fig. 3.

with unit cell parameters ranging from 11.108 (2)–11.163 (3) Å, which are in good agreement with those calculated from the powder XRD data and those previously reported[13]. As shown in Fig. 5a–b, the Ag$_9$GaSe$_6$ structure features a rigid anion framework of [GaSe$_6$]$^{9-}$ that is embedded with tetrahedral GaSe$_4$ (formed merely by the Se1 atoms occupying the 16$e$ site, Supplementary Tables 4–6). Single crystals selected from the same batch of samples show great consistency with an atomic coordination fluctuation less than 5‰ of the unit cell length (Supplementary Tables 4–6). During different condensation processes, neither the PXRD diffraction peaks of the Ag$_9$GaSe$_6$ samples

(Supplementary Fig. 5) nor the atomic coordinates of the Ga and Se atoms change detectably, which shows a high consistency with only a slight fluctuation of < 0.9‰ of the unit cell. However, such condensation processes promote never-discovered atomic migration of the Ag atoms. Supplementary Table 4 shows that in the as-synthesized Ag$_9$GaSe$_6$ crystal structure, there are four crystallographically independent Ag sites per unit cell located at different Wyckoff sites with different occupancies, e.g., Ag1 is threefold coordinated by Se2 and Se1 atoms and located at the 24 $g$ site with an occu. of 0.5, which also centers the triangle surface of the Se$_4$-tetrahedron. Ag2 is also threefold coordinated

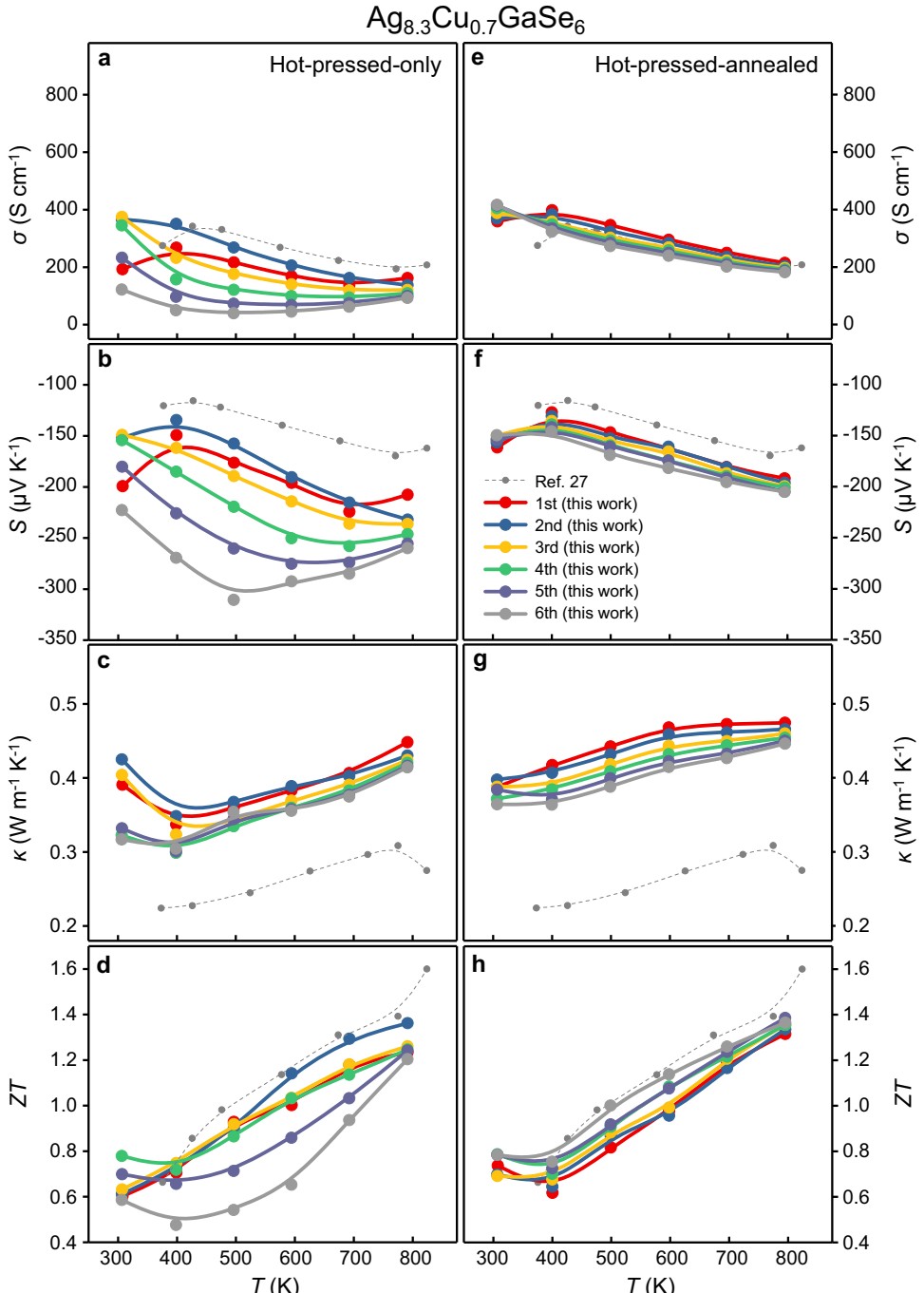

**Fig. 4 Temperature-dependent thermoelectric properties of $Ag_{8.3}Cu_{0.7}GaSe_6$. a–d** hot-pressed-only and **e–h** hot-pressed-annealed samples. **a**, **e** Electrical conductivity $\sigma$. **b**, **f** Seebeck coefficient $S$. **c**, **g** Thermal conductivity $\kappa$. **d**, **h** Figure of merit $ZT$. The Dulong-Petit-$C_p$ was used. Those reported in ref. [27] that are calculated by the Dulong-Petit-$C_p$ are presented as references. The $\kappa$ and $ZT$ calculated by the measured-$C_p$ are provided in Supplementary Fig. 4.

in a planar triangle manner and is located at site $24\,f$ with an occu. of 0.15. Ag3 occupies the $48\,h$ site with an occu. of 0.20 at the center of the triangle surface of a $Se_4$-tetrahedron. Ag4 at the $48\,h$ site with an occu. of 0.20 centers the $Se_4$-tetrahedron (Fig. 5). Interestingly, under external condensation treatment stimuli, Ag migrates via two routes:

(Route I) Reversible in-and-out migration from the triangle surface-center site (Ag1) to the body-center site (Ag5) of the tetrahedron. The Ag1–Se bond length hardly changes during different condensation processes (Table 1). However, the occupancy declines from 0.5 (of the as-synthesized crystals,

Supplementary Table 4) to 0.35 (Supplementary Table 5) during hot pressing. Whereas after the subsequent annealing, the occupancy increases back to the original value of 0.5. These results infer that upon exposure to high press stimuli during the hot-pressing treatment, approximately 15% of the Ag1 atoms shift from the 3-fold coordinated triangular-surface-center to the 4-fold coordinated body-center in the $Se_4$ tetrahedron.

(Route II) Shifting of the Ag2 atom towards the planar triangle center. The occupancy variation on the Ag2 site during different condensation treatments changes less obviously. However, judging from the variation in the Ag2–Se bond length, a shift

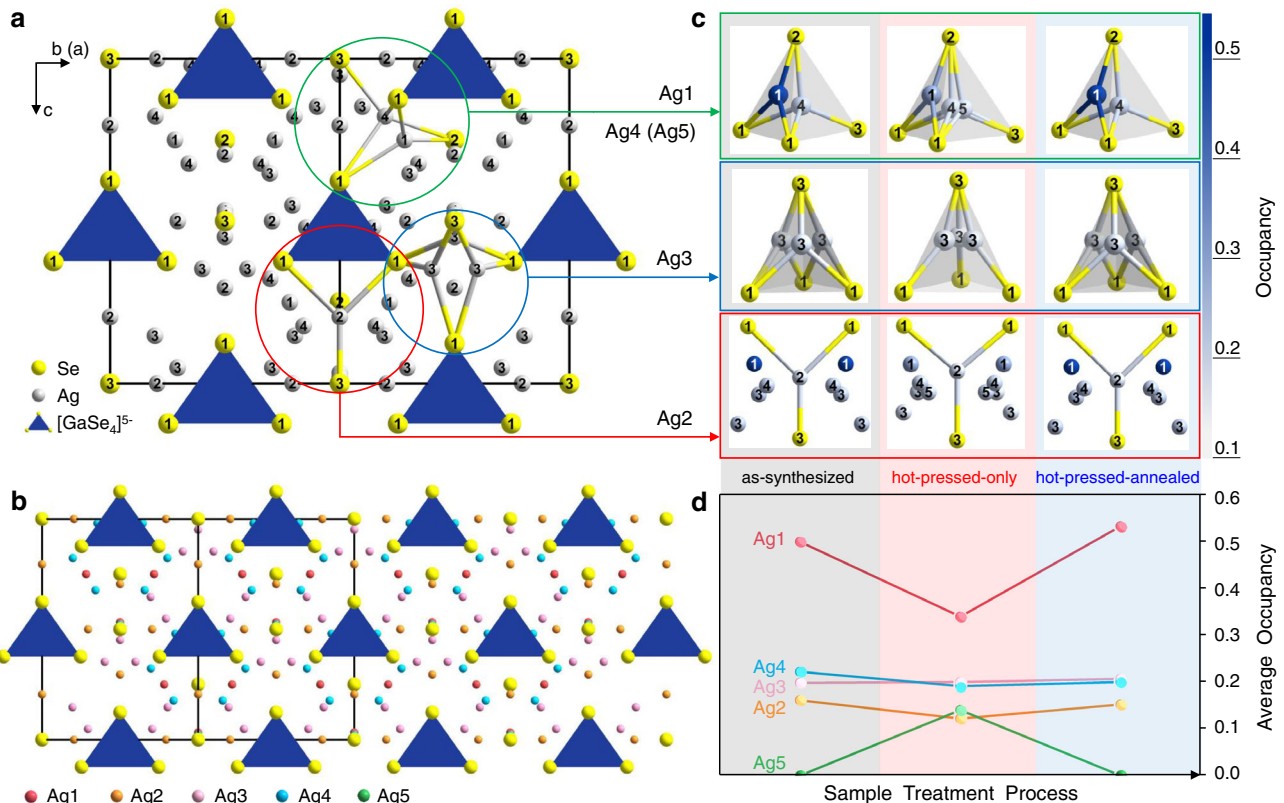

**Fig. 5 Single crystal structure of Ag$_9$GaSe$_6$, with the Ag local coordination environment and site occupancy obtained by the single crystal diffraction data. a** The unit cell with atoms numbered. Selected Ag–Se bonds are shown. **b** Structure view with the unit cell marked. The crystallographic independent Ag atoms are marked by different colors. **c** The Ag local coordination of all crystallographic independent Ag1–5 atoms in a unit cell. The right-side color code indicates the Ag site occupancy. **d** The occupancy change as a function of the sample treatment. Each average occupancy is obtained by the arithmetic average of those refined by 9 different single crystals selected from each batch of the samples. Detailed crystallographic data, refinements, atomic coordinates, anisotropic displacement parameters and occupancies are listed in Supplementary Tables 1–12 in the Supplementary Information.

**Table 1 Selected bond distance (Å) of single crystals obtained by as-synthesized, hot-pressed-only, and hot-pressed-annealed process.**

|  | As-synthesized | Hot-pressed-only | Hot-presssed-annealed |
|---|---|---|---|
| Ag1–Se1 | 2.64 | 2.66 | 2.63 |
| Ag1–Se2 | 2.42 | 2.39 | 2.44 |
| Ag2–Se1 | 2.72 | 2.55 | 2.71 |
| Ag2–Se3 | 2.28 | 2.54 | 2.30 |
| Ag3–Se1 (nearest) | 2.70 | 2.49 | 2.73 |
| Ag3–Se1 (next-nearest) | 3.67 | 3.22 | 3.75 |
| Ag3–Se3 | 2.37 | 2.27 | 2.35 |
| Ag4–Se1 | 2.80 | 2.70 | 2.82 |
| Ag4–Se2 | 2.81 | 2.79 | 2.80 |
| Ag4–Se3 | 2.80 | 2.96 | 2.78 |
| Ag5–Se1 | — | 2.97 | — |
| Ag5–Se2 | — | 2.89 | — |
| Ag5–Se3 | — | 2.50 | — |

Notes: These distances are obtained by the #1 crystal of the corresponding batch of samples. The extended data are provided in Supplementary Tables 1–9 in Supplementary Information.

towards the center of the Se$_3$ triangle plane is clearly suggested. Table 1 shows that at high pressure, the Ag2–Se3 bond increases from 2.28 to 2.54 Å; simultaneously, Ag2–Se1 shortens from 2.72 to 2.55 Å, indicating movement towards the center of the Se$_3$-triangle. As shown in Fig. 5c bottom, such shift is driven towards the triangle center by an increase in the static electric repulsion owing to the emergence of the Ag5 atoms and a decrease in the repulsion from Ag1 atoms because of its 15% moving-out migration, as described by route I (Fig. 5c).

In comparison, the migration of Ag3 and Ag4 is a minor event, and their occupancies are nearly constant at different treatments (Fig. 5d). Ag3 is located at the center of the three side surfaces of the tetrahedron. Under high pressure, Ag3 slightly moves towards the body center of the tetrahedron. The Ag4 located at the tetrahedron body center is more stationary, and the Ag-Se bond length does not vary noticeably.

To further establish the correlation between the Ag migration and the hot-press pressure, the hot-pressing procedures were carried out under different pressures (30–90 MPa), during which high quality small single crystals were obtained. (Supplementary Tables 10–12) Together with the high quality small single crystals that were as-grown inside the corresponding ingots (Supplementary Tables 1–9), we confirmed the pressure-dependent Ag migration routes as revealed by the occupancy variation of the corresponding Ag atoms. (Supplementary Fig. 8) When the pressure is below 60 MPa, the Ag occupancy is nearly constant, indicating the relative stationaries of all the Ag atoms; when the pressure increases beyond 60 MPa, the population of the 4-fold coordinated Ag atoms increases, which equals to the decrease amount of the 3-fold-coordinated Ag atoms. Such a variation proves straightforwardly the Ag-migration driven by the high pressure, which on the other hand illustrates that Ag atoms tend

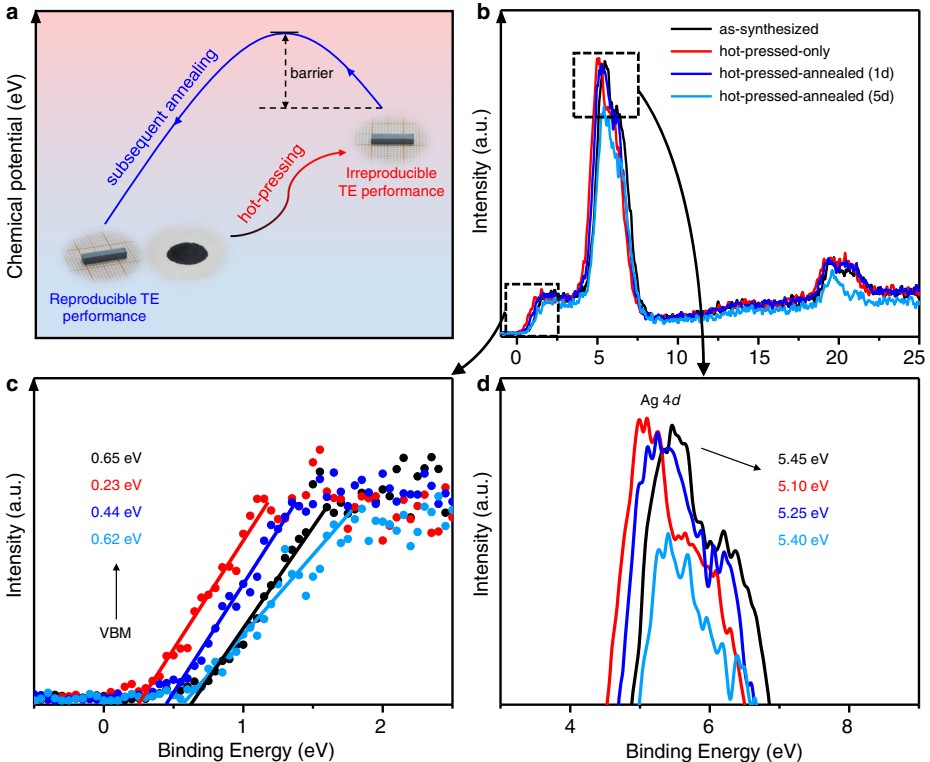

**Fig. 6 Chemical potential of Ag$_9$GaSe$_6$ samples. a** Sketch illustration of the change of the chemical potential during the sample condensation process. **b** VBXPS spectra of the as-synthesized, hot-pressed-only and hot-pressed-annealed samples. For the latter, samples annealed for 1 or 5 day(s) are presented. **c** The edge of the valence band maximum and **d** the characteristic peak of the Ag 4$d$ orbital.

to adopt a higher-coordination environment under high pressure. Similar phenomena were observed by the Xe–F coordination number increasing from 2 to 4 under 23 GPa, then to 8 under 70 GPa[46], as well as by the Cs–F coordination number increasing from 1 to 6 under high pressure[47].

Indeed, the hot-pressed-only sample has high chemical potential, as suggested by the binding energy measured by valence band X-ray photoelectron spectroscopy (VBXPS) (Fig. 6). The edge of the valence band maximum (VBM) energy of the hot-pressed-only Ag$_9$GaSe$_6$ sample is 0.23 eV below the Fermi level, much higher than the 0.65 eV of the as-synthesized sample. By subsequent annealing for 1 day, the VBM of the hot-pressed-annealed sample decreases to 0.44 eV; furthermore, after 5 days of annealing, the VBM decreases to 0.62 eV, nearly returning to that of the original as-synthesized sample. In addition, the binding energy of the Ag 4$d$ orbital[48,49] is as high as 5.45 eV in the as-synthesized sample and decreases to 5.10 eV after hot-press condensation. Such a decrease indicates that the valence electrons of Ag ions are more easily excited by the incident photons, indicating a higher intrinsic chemical potential and less stable state. The binding energy of the Ag 4$d$ orbital increases to 5.25 eV after annealing treatment for 1 day and further increases to 5.40 eV after annealing for 5 days. These data sufficiently prove that the hot-pressed-only sample has a lower binding energy and thus a higher chemical potential that leads to an unstable state that is manifested in the instability and TE property irreproducibility, as presented above. To eliminate such instability, an energy barrier needs to be overcome, which is partially overcome after 1 day of annealing and completely overcome after 5 days of annealing treatment (Fig. 6a).

To summarize, as shown in Fig. 6a, during hot-pressing condensation, high pressure induces the liquid-like Ag atoms to migrate to a position with a higher coordination number that is also higher in chemical potential but lower in binding energy. Such high chemical potential Ag distribution inevitably leads to unwanted TE performance instability and irreproducibility. To eliminate such a high-pressure-induced high chemical potential state, an energy barrier has to be overcome, which is realized by a subsequent annealing at 823 K for at least 24 h that drives the high-energy metastable Ag atom distribution back to the original crystallographic sites with low chemical potentials. Remarkably, Ag$_9$GaSe$_6$ with Ag atoms distributed over low chemical potentials is proven to be intrinsically stable. In this study, a never observed before, great TE property reproducibility on both pristine and optimized Cu/Te-doped Ag$_9$GaSe$_6$ is realized.

## Discussion

In summary, we present an innovative discovery of the liquid-like high-performance TE material Ag$_9$GaSe$_6$ and its optimized derivatives, and the most unwanted property instability and irreproducibility are eliminable. We demonstrate that induced by high pressure, pure phase as-synthesized powdery polycrystalline Ag$_9$GaSe$_6$-based samples undergo silver migration during the hot-press procedure, giving rise to a metastable Ag distribution carrying a higher chemical potential and lower binding energy. With solid and convincing VBXPS and single crystal diffraction data, we show that such instability is eliminated by a simple subsequent annealing treatment at 823 K for at least 24 h that efficiently drives the metastable Ag atoms to migrate back to the original energetically more stable crystallographic sites. Furthermore, we show that the annealed samples exhibit great stability and repeatability of the TE properties; for instance, hot-pressed-annealed Cu/Te-doped Ag$_9$GaSe$_6$ constantly measures a $ZT = \sim 1.4$ at 800 K after multiple measurements, which is one of the highest ever reported values for this family, and this result has never been previously

reported. This discovery will prompt the scientific community to reconsider argyrodite and other related liquid-like materials in terms of the long-believed misunderstanding of their instability and performance irreproducibility; in addition, their application potential is worth re-evaluating.

## Methods

**Sample preparation**. All samples were prepared by solid-state reactions and condensed by the hot-pressing technique accordingly[16,17,19,20,25]. Ag (powder), Ga (shots), Se (powder), Te (powder), and Cu (powder) with purities above 99.999% were purchased from Alfa Aesar and used as received. The stoichiometric reactants were weighed inside an argon-filled glove box and then loaded into silica tubing. Then, the reactant was sealed under vacuum with a residual pressure of less than $10^{-3}$ Pa. The assembly was heated to 1373 K within 12 h and held for 12 h before quenching in cold water. The quenched ingots were annealed at 873 K for 3 days (denoted as as-synthesized sample). The obtained products were ground into fine powders that were subsequently condensed by either a hot-pressed-only process at 873 K under a pressure of 60 MPa for 1 h (denoted as hot-pressed-only sample); or by the same hot-pressed process followed by annealing at 823 K for 24 h under a $N_2$ flow (denoted as hot-pressed-annealed sample). The hot-pressed-only and hot-pressed-annealed samples were well condensed to a high density great than 97%, as listed in Supplementary Table 13.

To pick up the suitable single crystals for the crystallographic structure determination, the above-mentioned as-synthesized ingots were crushed into pieces, within which a large number of high-quality small single crystals are obtained. These small crystals are black and shining, with sizes about tens of microns. (Supplementary Fig. 1) The suitable single crystals were then hand-picked, and used to collect the single-crystal diffraction data. (Supplementary Tables 1–12)

According to the scanning electron microscopy (SEM) observations on the hot-pressed-annealed samples, no obvious metal precipitation or Se violation were detected. (Supplementary Fig. 9) The corresponding powder XRD and single-crystal diffraction data also indicate the consistent homogenous phase composition.

**Characterization**. A Bruker D8 ADVANCE equipped with Cu $K_\alpha$ radiation was used to collect the powder X-ray diffraction (XRD) data, and silicon powder (99.99%, Aladdin) was added as an internal standard to ensure the accuracy of the cell parameter calculation. Single-crystal diffraction data were collected on a Bruker APEX-II CCD equipped with Mo $K_\alpha$ radiation. The structure determination was based on a full-matrix least-square refinement on $F^2$ using the SHELXTL97 program package[50]. The valence band X-ray photoelectron spectroscopy (VBXPS) measurements were performed on a Thermo Scientific ESCALAB 250Xi instrument with an Al $K_\alpha$ source. The morphology of samples was characterized by a field emission scanning electron microscopy (FESEM, S-8010, Hitachi) equipped with Energy Dispersive Spectrometer (EDS, XFlash6160, Bruker).

**Electrical and thermal transport property measurements**. The Seebeck coefficient and electrical conductivity were measured synchronously using an Ulvac Riko ZEM−3 instrument. The total thermal conductivity was calculated from $\kappa = DC_p d$, where $D$ is the thermal diffusivity measured on Netzsch LFA-457; the heat capacity $(C_p)$ is measured on a Netzsch STA449 F5 and is also estimated by the Dulong-Petit limit, and $d$ is the density of the sample measured by the Archimedes method. The measurement uncertainties for $S$, $\sigma$ and $\kappa$ are ~ 5%. Note that all the property measurements were executed below 800 K to protect the facilities since at the higher temperature the samples may undergo Se volatilization which will contaminate the thermocouple probe. More details regarding the critical voltage determination can be found in Supplementary Information.

## Data availability

Supplementary Information and all single-crystal XRD data accompany this paper at http://www.nature.com/ naturecommunications. The single-crystal XRD crystallographic data reported in this study have been deposited in the Cambridge Crystallographic Data Center (CCDC) under deposition numbers of 2170787–2170829. These data can be obtained free of charge from the Cambridge Crystallographic Data Centre via http://www.ccdc.cam.ac.uk/ data_request/cif.

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

## Acknowledgements

This research was supported by the National Natural Science Foundation of China under projects (22193043, 21975032) received by L. C. We thank prof. P. -F. Qiu and Dr. Z. -C. Jin. (Shanghai Institute of Ceramics, Chinese Academy of Sciences) for the critical voltage measurement and valuable discussions.

## Author contributions

L.C. and L.-M.W. conceived and designed the experiments. J.-Y.L. synthesized the samples and performed all the experiments. The manuscript was written through contributions of all authors.

## Competing interests

The authors declare no competing interests.
