## [Peer review file · Nature Communications]

REVIEWER COMMENTS

Reviewer #1 (Remarks to the Author):

In this manuscript, Liu et al. investigated the chemical stability of a typical liquid-like thermoelectric material, argyrodite Ag_9GaSe_6 , using a different heat treatment in combination with valence band X-ray photoelectron spectra and refined data from X-ray diffraction. They showed the instability of the normally hot-pressed samples can be eliminated by an extra simple annealing treatment, which can drive the metastable Ag ions back to the more stable atomic sites. The manuscript is well written and the results are also very interesting. A major revision is needed before publication in Nature Communications.

1. Why did the authors state that ‘the p-type TE material is superior to its n-type counterpart in performance’? The given examples of P-type Cu_2Q and N-type Ag_9GaSe_6 are not much of the same class materials.
2. The authors claimed the Ag migration was caused by the high pressure, but the evidences are not solid enough. I suggest pressing the samples at different pressures.
3. Why atoms tend to adopt a higher-coordination environment under high pressure?
4. How did the authors grow the single crystals? I also don’t understand how to get single crystals after hot pressing or hot-pressing-annealing processes.
5. Ag_9GaSe_6 is a n-type semiconductor, why did the authors measure the edge of the valence band maximum? What is valence band X-ray photoelectron spectroscopy (VBXPS)? Is this method really able to derive the information about valence band maximum? It seems the VBM energy change a lot after annealing, why?

Reviewer #2 (Remarks to the Author):

The manuscript entitled “ Ag_9GaSe_6 : High-pressure-induced Ag migration causes thermoelectric performance irreproducibility and elimination of such instability” by Li-Ming Wu and co-workers demonstrate the effect of annealing treatment on the reproducibility of thermoelectric data. As per the authors claim annealing treatment promotes the Ag-atoms to the more stable low chemical potential site and inhibits high-pressure induced Ag-migration after the hot-press. The work lacks providing novel fundamental ideas as it is quite well known in the literature that annealing treatment in most of the cases reduces the irreversibility in both thermal and electrical transport properties. Also, after the thermal annealing there is no overall improvement in thermoelectric performance of Ag_9GaSe_6 .

Therefore, I feel the manuscript is not suitable for such a high-impact journal, Nature Communication. Also, there are several other concern authors must take care.

1. Authors have claimed that they have collected single crystal diffraction data of as-prepared, hot-pressed-only, and hot-pressed-annealed samples. But how the authors have obtained single crystal just after hot pressing the powder sample or after the annealing of the hot-pressed sample? Also, authors have not given any single crystal X-ray diffraction pattern.

2. Authors have also mentioned that 15% of the Ag1 atoms shift from the 3-fold coordination to the 4-fold coordination environment whereas occupancy variation on the Ag2 site during different condensation treatments changes is less. Then it is expected that thermal displacement parameter will be high for Ag1 than Ag2. But the refinement data of both as-synthesized single crystal and hot-pressed-only crystal (in the supplementary information) indicates the reverse fact (for hot-pressed-only crystal $U_{eq} = 0.104$ (Ag1) and 0.227 (Ag2))

3. Also, only isotropic thermal displacement parameters are provided. Do all the atoms behave isotropically?

4. For many chalcogenide systems it is reported that prolonged annealing leads to volatilization of S or Se (ACS Appl. Mater. Interfaces 2021, 13, 45736–45743). Authors must perform the microscopy to find out the exact composition of the samples after annealing.

5. Authors must mention the density of the samples in comparison to the theoretical density.

6. It is recommended to use the measured C_p for the estimation of thermal conductivity and zT , which will reduce the error in zT estimation. Also, authors have compared their data with ref. 15 where Chen et al. have used the measured C_p (higher than Dulong-Petit C_p) for thermal conductivity and zT estimation of Ag_9GaSe_6 . During comparison authors must maintain similar experimental parameters.

Reviewer 3 comments:

This paper presents some results about the improved stability and reproducibility of argyrodite Ag_9GaSe_6 via annealing treatment. The authors claim that high pressure promotes the migration of Ag-sublattice to sites with high chemical potentials, and annealing would drive the metastable Ag atoms to migrate back to the original energetically more stable crystallographic sites. The finding is interesting, and I thus recommend major revision based on the following reasons.

1. This paper only shows the transport properties until 800 K. However, the important references cited in this paper studied the thermoelectric properties up to 823 K. It is necessary to do the comparison only if you give the thermoelectric data and reproducibility until 850 K.
2. How about the stability related to the current density? The stability depends on the applied electronic field should be provided, especially the critical voltage.
3. The theoretical calculation should be carried out to clearly show the migration paths of Ag ions.
4. TEM are strongly recommended to observe the possible precipitation of Ag nanoparticle.

Point-by-point answers

Reviewer #1:

In this manuscript, Liu et al. investigated the chemical stability of a typical liquid-like thermoelectric material, argyrodite Ag₉GaSe₆, using a different heat treatment in combination with valence band X-ray photoelectron spectra and refined data from X-ray diffraction. They showed the instability of the normally hot-pressed samples can be eliminated by an extra simple annealing treatment, which can drive the metastable Ag ions back to the more stable atomic sites. The manuscript is well written and the results are also very interesting. A major revision is needed before publication in Nature Communications.

Response: Thank you very much for the constructive, insightful and supportive comments of our work.

1. Why did the authors state that ‘the p-type TE material is superior to its n-type counterpart in performance’? The given examples of P-type Cu₂Q and N-type Ag₉GaSe₆ are not much of the same class materials.

Response: Our original idea is to express that the *p*-type **liquid-like** TE material is superior to its *n*-type counterpart in performance. In the revision, we have added ‘liquid-like’ to avoid the ambiguity.

Faithfully following your suggestions, the following sentences are added on p. 2, l.9 in the main text: “As listed in Table S14, the typical *p*-type liquid-like TE materials (such as the Cu₂Q-based compounds⁵, Cu-argyrodite^{21–25}, and superionic conductors^{29–36}) are more abundant than the *n*-type counterparts (e.g., Ag₂Q-based compounds,^{37,38} Ag-argyrodite.^{14–20, 26–28}) Besides, the best *p*-type liquid-like TE material can realizes a *ZT* of 2.6 in Cu₂Se/CuInSe₂,³⁹ whereas the best *n*-type one can only reaches a *ZT* of 1.6 by Ag₉GaSe₆.^{16, 26, 27} Thus, to explore on the *n*-type liquid-like materials are of great significance.”

New Table S14. Typical *p*-type and *n*-type liquid-like thermoelectric materials.

Type	Material	ZT _{max}	Structure type	Type	Material	ZT _{max}	Structure type	
p	Cu _{1.8} S + 0.75% graphene ⁵²	1.5 at 873K	Cu ₂ Q	n	Ag ₂ S ⁵¹¹	0.55 at 580 K	argyrodite	
	Cu ₂ S + 2 mol% In ₂ S ₃ ⁵³	1.2 at 850 K			Ag ₂ Se ³⁷	1.2 at 390 K		Ag ₂ Q
	Cu _{1.97} S ⁹	1.7 at 1000 K			Ag ₂ Te + PbTe + Ag ³⁸	1.0 at 500–600K		
	Cu ₂ Se ²	1.6 at 1000 K			Ag ₉ GaSe ₆ ^{15,16,26,27}	1.6 at 850 K		
	Cu ₂ Se + 0.75% CNTs ⁵⁴	2.4 at 1000 K			Ag ₉ GaS _{5.4} Se ₆ ¹⁴	0.6 at 800 K		
	Cu ₂ Se + 1 mol% In ³⁹	2.6 at 850 K			Ag ₉ Al _{0.06} Cd _{0.04} Se ₆ ¹⁷	1.0 at 850 K		
	Cu ₂ Se _{0.92} S _{0.08} ⁵⁵	2.0 at 1000 K			Ag ₉ GeSe _{5.88} ¹⁸	0.55 at 923 K		
	Cu ₂ Se + 0.6% graphite ⁵⁶	2.4 at 850 K			Ag ₉ Sn _{1-x} Nb _x Se ₆ (x≤0.05) ³⁹	1.2 at 850 K		
	Cu ₂ Te ⁵⁷	1.1 at 1000 K						
	Cu ₂ S _{0.52} Te _{0.48} ⁵⁸	2.2 at 1000 K						
	Cu _{1.98} S _{1/3} Se _{1/3} Te _{1/3} ⁵⁹	1.9 at 1000 K						
	Cu ₂ Se _{0.7} Te _{0.3} ⁵¹⁰	1.4 at 1000 K						
	Cu ₂ PSe ₄ ²⁴	0.35 at 575 K			argyrodite			
	Cu _{7.6} Ag _{0.4} GeSe _{5.1} Te _{0.9} ²³	1.0 at 800 K						
	Cu ₅ FeS ₄ + Cu ₂ S ²⁹	1.2 at 900 K						
	Cu _{4.972} Fe _{0.968} S ₄ ³⁰	0.84 at 675 K	Cu ₅ FeS ₄					
	Cu ₅ FeS ₄ ²¹	0.62 at 710 K						
	CuAgSe ¹²	0.95 at 623 K	CuAgSe					
	CuAgSe ¹³	0.9 at 623 K						
	Ag _{0.96} CrSe ₂ ²⁴	0.5 at 723 K						
CuCrSe ₂ ²⁵	1.0 at 773 K	AgCrSe ₂						
(AgCrSe ₂) _{0.5} (CuCrSe ₂) _{0.5} ³⁶	1.4 at 773 K							
Ag ₈ GeTe ₆ ²¹	0.48 at 703 K	argyrodite						
Ag ₈ SiTe ₆ ²²	0.48 at 800 K							
Ag ₉ Ga _{0.95} Cd _{0.05} Te ₆ ²⁵	0.65 at 600 K							

2. The authors claimed the Ag migration was caused by the high pressure, but the evidences are not solid enough. I suggest pressing the samples at different pressures.

Response: Thank you for your comments. In the revision, we have supplied 16 sets of single crystal diffraction data on samples that are obtained under various pressures: 0, 30, 40, 50, 52.5, 55, 57.5 and 90 MPa. (listed in new Tables S10–12 in the Supporting Information) Note that the 90 MPa is the accurate pressure threshold of our hot-pressing furnace. Altogether, the 43 sets of data demonstrate a trend in the Ag_9GaSe_6 unit cell, in which when the pressure is below 60MPa, all Ag atoms remain relative constant occupancies that are irrelevant to the pressure indicating the Ag migration is not initiated yet. (Fig. S8a) As the pressure increases beyond 60 MPa (note that 60MPa is the required pressure to condense the sample during the standard hot-press process), the Ag atoms tend to migrate away from the 3-fold coordinated sites to the 4-fold coordinated sites. As Fig. S8 shown, the occupancies of Ag1 and Ag3 begin to decrease, and those of Ag4 and Ag5 start to increase, whereas that of Ag2 remains almost unchanged. Eventually, under pressure higher than 60MPa, the population of the 3-fold-coordinated Ag atoms decreases, whereas that of the 4-fold-coordinated Ag atoms increases. (Fig. S8) What is more illustrative is that the net decrease number of the 3-fold-coordinated-Ag atoms exactly equals to the net increase of the 4-fold-coordinated-Ag, which means that the 3-fold-coordinated Ag atoms begin to migrate to the 4-fold-coordinated sites when the pressure is higher than 60MPa. In another word, under higher pressure, Ag tends to adopt a higher coordination number. Similar phenomena were observed by the Xe–F coordination number increasing from 2 to 4 under 23 GPa, then to 8 under 70 GPa,⁴⁶ (*Nat. Rev. Mater.* **2010**, 2, 784); as well as by the Cs–F coordination number increasing from 1 to 6 under high pressure.⁴⁷ (*Nat. Chem.* **2013**, 5, 846.)

Faithfully following your suggestions, the following sentences are added on p.7, l.33: "... ... To further establish the correlation between the Ag migration and the hot-press pressure, the hot-pressing procedures were carried out under different pressures (30–90 MPa), during which high quality small single crystals were obtained. (Tables S10–12) Together with the high quality small single crystals that were as-grown inside the corresponding ingots (Tables S1–9), we confirmed the pressure-dependent Ag migration routes that are revealed by the occupancy variation of the corresponding Ag atoms. (Fig. S8) When the pressure is below 60 MPa, the Ag occupancy is nearly constant, indicating the relative stationaries of all the Ag atoms; when the pressure increases beyond 60 MPa, the population of the 4-fold coordinated Ag atoms increases, which equals to the decrease amount of the 3-fold-coordinated Ag atoms. Such a variation proves straightforwardly the Ag-migration driven by the high pressure, which on the other hand illustrates that Ag atoms tend to adopt a higher-coordination environment under high pressure. Similar phenomena were observed by the Xe–F coordination number increasing from 2 to 4 under 23 GPa, then to 8 under 70 GPa,⁴⁶ as well as by the Cs–F coordination number increasing from 1 to 6 under high pressure.⁴⁷"

The new Fig. S8 and new Tables S10–12 are supplied in the Supporting Information.

Fig. S8. Hot-press pressure-dependent **a)** average occupancy of each Ag atom and **b)** total numbers per unit cell of the 3-fold- (Ag1 + Ag2 + Ag3) and 4-fold-coordinated (Ag4 + Ag5) Ag atoms, respectively. Points are experimental data listed in Tables S10–12 and the dotted line is the fitting curve.

3. Why atoms tend to adopt a higher-coordination environment under high pressure?

Response: Thanks. See also the answers in **R1**, **Q2** above. Here, we also explain this issue in a more general way as below. It's a common phenomenon that atoms tend to adopt a higher-coordination environment under high pressure. The most straightforward effect of pressure on materials is the volume decrease, which leads to a reduction of the interatomic distances and the space around atoms. (*Nat. Rev. Mater.* **2017**, 2, 17005) Once the atoms are compressed, there is a tendency to migrate to larger space. Normally, site with higher-coordination numbers always has larger surrounding spaces than the one with lower-coordination numbers, thus, under high pressure atoms tend to adopt a higher-coordination environment. For example, some binaries (e.g., AlN, AlP, AlAs; GaN; InN, InP, InAs) and ZnQ, CdQ (Q = S, Se, Te) transform from the 4-fold coordinated zincblende or wurtzite structures to the 6-fold coordinated NiAs or NaCl structures under high pressure. (*Rev. Mod. Phys.* **2003**, 75, 863) The Xe-F coordination number increases from 2 to 4 (~ 23 GPa) then to 8 (~ 70 GPa). (*Nat. Chem.* **2010**, 2, 784) The 4-fold coordinated germanium in liquid germanate shows an abrupt change to the 6-fold coordination around 3 GPa (*Phys. Rev. Lett.* **2004**, 92, 155506); In GaAsO₄, the 4-fold coordinated Ga and As transform to the 6-fold coordination under pressure (*Europhys. Lett.* **1997**, 40, 533), so does AlPO₄ (*Nat. Mater.* **2007**, 6, 698); the 3-fold-coordinated B atoms in Li₂B₄O₇ change to the 4-fold coordination at around 5 GPa and the population of the 4-fold coordinated B atoms increases from 50% (at 1 atm) to more than 95% (at 30 GPa) (*Phys. Rev. Lett.* **2007**, 98, 105502).

4. How did the authors grow the single crystals? I also don't understand how to get single crystals after hot pressing or hot-pressing-annealing processes.

Response: Thanks a lot. The single crystals reported herein are not the large-sized single-crystals grown by a special crystal-growth method, rather, they are as-grown inside the as-synthesized ingots. When we crushed the as-synthesized ingots into pieces, a large number of tiny crystals with sizes of about tens of microns are observed. Then, the suitable single crystals were hand-picked under a microscope and used to collect the single crystal diffraction data.

Following your suggestions, the following sentences are added in the "Sample preparation" section under Methods on P. 9, l.13–20, "... ..To pick up the suitable single crystals for the crystallographic structure determination, the above-mentioned as-synthesized ingots were crushed into pieces, within which a large number of high-quality small single crystals are obtained. These small crystals are black and shining, with sizes about tens of microns.(Fig. S1) The suitable single crystals were then hand-picked, and used to collect the single-crystal diffraction data. (Tables S1–12) "

A new Fig. S1 is supplied (copied below).

Fig. S1. a, b, The as-synthesized Ag_9GaSe_6 ingot product. c The as-synthesized ingot was crushed and small single crystals (several are marked in yellow) are picked to collect the single crystal diffraction data. d The crystal with a size of tens of microns (marked in red) is set on the loop. e A screenshot image of the clear and orderly diffractions during the single crystal diffraction data collection. f In order to display the diffractions more clearly, we display the original diffraction pattern in grayscale.

5. Ag₉GaSe₆ is a n-type semiconductor, why did the authors measure the edge of the valence band maximum? What is valence band X-ray photoelectron spectroscopy (VBXPS)? Is this method really able to derive the information about valence band maximum? It seems the VBM energy change a lot after annealing, why?

Response: The VBXPS is actually X-ray photoelectron spectroscopy, but it stresses the low binding energy region (usually 0 to tens eV) close to the Fermi level. Electrons excited by photons of such low energy are no longer the core level electrons, but valence electrons. In general, due to the valence electron interactions of atoms in the solid and the limitation of the line width of the X-ray source, the lines of the valence band spectrum will be close together, and a broad plateau will appear near the Fermi level on the spectrum (e.g., the region of 1.5–4 eV in Fig. 6b), which reflects the electronic structure information of a solid.

By the XPS measurement data, the valence band maximum (VBM) is determined by the intersection of the linearly fitting of the leading edge of the valence band and the flat energy baseline (*Phys. Rev. Lett.* **1980**, *44*, 1620). The VBM represents the difference between the valence band edge energy and the Fermi level, which has nothing to do with the semiconductor nature whether it is a p-type or an n-type. For instance, Chen et al. used the same VBXPS technique to indicate a VBM blue-shifts toward the vacuum level observed in the black TiO₂ nanocrystals in comparison with that (at – 0.92 eV) of a white TiO₂ (*Science* **2011**, *331*, 746). Similarly, Kanatzidis et al. measured the VBMs of p-type AgMnSbTe₃ and Ag₂Te (*J. Am. Chem. Soc.* **2021**, *143*, 13990).

Meanwhile, such a method is also widely used for the n-type materials. Literature reports that with/without the F-doping, the VBM values of n-type TiO₂ films are the same indicating that the F-doping does not affect the density of states in the valence band (*J. Mater. Chem. A* **2014**, *2*, 3513); in the n-type InGaAs nanowires, the separation from the Fermi level to the surface VBM verifies a composition-dependent band bending that is associated the increase of the Ga-content (*Nano Lett.* **2016**, *16*, 5135–5142); even, the conduction type conversion from n- to p-type is detected by the VBM change after doping Zn in α -Fe₂O₃ (*J. Appl. Electrochem.* **2021**, *51*, 521).

In our work, the VBM was measured to compare the energy differences among the hot-pressed-only, hot-pressed-annealed (1 day) and hot-pressed-annealed (5 days) samples. The principle of XPS is to irradiate the sample with the X-rays so that the inner electrons or the valence electrons are excited and emitted. The low binding energy indicates the corresponding electron is easily to be excited, that is, the electron itself has high energy and is relatively unstable. Herein, we take the advantage of this point to indirectly compare the stability of the valence electrons by comparing the corresponding VBM binding energy, and eventually to illustrate the sample stability.

Regarding your last concerns on “the VBM energy change a lot after annealing”, the reason is that after annealing, the Ag atoms migrate back to the 3-coordinated sites that are thermodynamically more stable, (as shown in Fig. S8) The higher population of the 3-coordinated Ag atom that are more thermodynamically stable in the structure causes the higher excited energy threshold of the valence electrons.

Reviewer #2:

The manuscript entitled “Ag₉GaSe₆: High-pressure-induced Ag migration causes thermoelectric performance irreproducibility and elimination of such instability” by Li-Ming Wu and co-workers demonstrate the effect of annealing treatment on the reproducibility of thermoelectric data. As per the authors claim annealing treatment promotes the Ag-atoms to the more stable low chemical potential site and inhibits high-pressure induced Ag-migration after the hot-press. The work lacks providing novel fundamental ideas as it is quite well known in the literature that annealing treatment in most of the cases reduces the irreversibility in both thermal and electrical transport properties. Also, after the thermal annealing there is no overall improvement in thermoelectric performance of Ag₉GaSe₆. Therefore, I feel the manuscript is not suitable for such a high-impact journal, Nature Communication. Also, there are several other concern authors must take care.

Response: Thank you very much for the insightful evaluation and sincere advice of this work.

1. Authors have claimed that they have collected single crystal diffraction data of as-prepared, hot-pressed-only, and hot-pressed-annealed samples. But how the authors have obtained single crystal just after hot pressing the powder sample or after the annealing of the hot-pressed sample? Also, authors have not given any single crystal X-ray diffraction pattern.

Response: Please see answers to R1, Q4. The detailed descriptions are provided in the “sample preparation” section on p.9. Thanks.

2. Authors have also mentioned that 15% of the Ag1 atoms shift from the 3-fold coordination to the 4-fold coordination environment whereas occupancy variation on the Ag2 site during different condensation treatments changes is less. Then it is expected that thermal displacement parameter will be high for Ag1 than Ag2. But the refinement data of both as-synthesized single crystal and hot-pressed-only crystal (in the supplementary information) indicates the reverse fact (for hot-pressed-only crystal $U_{eq} = 0.104$ (Ag1) and 0.227 (Ag2))

Response: Thanks for your concerns. In fact, there is no direct relationship between the atom thermal vibration displacement parameter (U_{eq}) and the possible site that an Ag atom prefers to migrate to under high pressure. If the thermal vibration mode is like simple harmonic vibration, no matter how large the vibration amplitude is, the atoms will always vibrate around the equilibrium position that is lower in energy, and will not migrate to the other sites having higher energy. Just like a spring, whether being stretched or compressed, it will eventually return to the origin position ($\Delta x = 0$) with the lowest elastic potential energy. However, the Ag-migration discussed herein is not a simple harmonic vibration, the Ag atoms don't return to their original sites after leaving, but stay at the destination site of the migration which shows lower potential energy. Therefore, the U_{eq} , representing the magnitude of the thermal vibration amplitude, is not directly related to the tendency of the Ag-migration.

3. Also, only isotropic thermal displacement parameters are provided. Do all the atoms behave iso-tropically?

Response: Yes, all atoms are refined anisotropically, as indicated in the cif files. Following your suggestions, the anisotropic displacement parameters, including U_{11} , U_{22} , U_{33} , U_{12} , U_{13} and U_{23} for all atoms are provided in Supporting Information p 25–31 as Tables S7–9, and S12.

4. For many chalcogenide systems it is reported that prolonged annealing leads to volatilization of S or Se (ACS Appl. Mater. Interfaces 2021, 13, 45736–45743). Authors must perform the microscopy to find out the exact composition of the samples after annealing.

Response: Following your suggestions, we performed the scanning electron microscopy (SEM) on a polished Ag_9GaSe_6 pellet before and after the 1d-annealing at 823K (new Fig. S9 is supplied). Compared with what observed in the mentioned literature, no obvious Se-volatilization-induced pores, or metal precipitation were observed in our sample, even the scratches caused by sanding are still clearly visible, indicating the unaffected sample surface after annealing. The elemental distribution mappings on the right also proved that the composition is uniform and homogenous.

Besides, after the annealing treatment, the unit cell parameters calculated by the powder-XRD data merely change by a factor less than 0.09‰ in the a parameter (11.1436 vs. 11.1446 Å) and 0.3‰ in V (1383.80 vs. 1384.17 Å³) (Fig. S5b). These experimental observations strongly support that the phase composition is constant and stable under our experimental conditions.

Faithfully following your suggestions, the following sentences are added in the sample preparation as the 2nd paragraph on p. 9, 1.22-27:”.....According to the scanning electron microscopy (SEM) observations on the hot-pressed-annealed samples, no obvious metal precipitation or Se violation were detected. (Fig. S9) The corresponding powder XRD and single-crystal diffraction data also indicate the consistent homogenous phase composition.”

SEM images and elemental distribution mappings of the polished surfaces of Ag_9GaSe_6 ingots: our work (left, Fig. S9), and the reported work (right, the mentioned literature of ACS Appl. Mater. Interfaces 2021, 13, 45736–45743).

5. Authors must mention the density of the samples in comparison to the theoretical density.

Response: Done as suggested. The density and the comparison to the theoretical density are provided in Supporting Information as Table S13. (copied below)

Table S13. Experimental density measured by Archimedes method and relative density before and after annealing.

Sample	Before/after annealing	Density* (g/cm ³)	Relative density
Ag ₉ GaSe ₆	before	7.16	98.1%
	after	7.18	98.3%
Ag ₉ GaSe _{5.5} Te _{0.5}	before	7.20	98.6%
	after	7.22	98.9%
Ag _{8.3} Cu _{0.7} GaSe ₆	before	7.07	96.9%
	after	7.08	97.0%

* The density measurement uncertainty is within 1%.

6. It is recommended to use the measured C_p for the estimation of thermal conductivity and zT , which will reduce the error in zT estimation. Also, authors have compared their data with ref. 15 where Chen et al. have used the measured C_p (higher than Dulong-Petit C_p) for thermal conductivity and zT estimation of Ag₉GaSe₆. During comparison authors must maintain similar experimental parameters.

Response: Done as suggested. We measured the C_p for all samples using a NETZSCH STA449 F5, good consistency was observed before and after the annealing treatment, a new Fig 2 and a new section “comparison for the measured and Dulong-Petit estimated C_p ” and new Figs. S2–4 are provided in the Supporting Information. As shown in Fig. S2, the measured C_p with an average of 0.29 J/g·K is about 10% higher than the Dulong-Petit estimated value of ~0.2635 J/g·K, which consists well with the measured C_p in ref. 15. Notably, the peak around 330K as reported in Ref. 27 indicating a phase transition in Ag_{8.3}Cu_{0.7}GaSe₆ sample is also observed.

The κ and ZT of Ag₉GaSe₆ calculated by both the measured- and Dulong-Petit- C_p are presented in Fig. 2. Due to the measured C_p is higher than the Dulong-Petit estimated value, the corresponding κ is also higher with an average difference about 10%. Accordingly, the overall ZT using the experimental measured- C_p (Fig. 2) is averagely about 10% lower than that utilizing Dulong-Petit- C_p .

Besides, for Ag₉GaSe_{5.5}Te_{0.5} (Fig. S3) and Ag_{8.3}Cu_{0.7}GaSe₆ (Fig. S4), data calculated by the Dulong-Petit C_p (refs. 16, 27) as well as by the measured C_p (this work) are presented. Except for few points with differences about 10%, the κ and ZT calculated by the measured- C_p are highly consistent with those calculated by Dulong-Petit- C_p , indicating the accuracy and reasonableness of the Dulong-Petit estimation and the stability and reliability of our C_p measurement results.

New Fig. 2 | c, g. Thermal conductivity κ . **d, h.** Figure of merit ZT . The κ and ZT reported in refs. 15 (calculated by the measured- C_p), and 16, 27 (calculated by the Dulong-Petit- C_p) are also presented as references. The κ and ZT calculated by the measured- and Dulong-Petit- C_p are both shown for better comparison. More details are provided in Supporting Information (Figs. S2–4).

Reviewer #3:

This paper presents some results about the improved stability and reproducibility of argyrodite Ag_9GaSe_6 via annealing treatment. The authors claim that high pressure promotes the migration of Ag-sublattice to sites with high chemical potentials, and annealing would drive the metastable Ag atoms to migrate back to the original energetically more stable crystallographic sites. The finding is interesting, and I thus recommend major revision based on the following reasons.

Response: Thank you for your great supporting.

1. This paper only shows the transport properties until 800 K. However, the important references cited in this paper studied the thermoelectric properties up to 823 K. It is necessary to do the comparison only if you give the thermoelectric data and reproducibility until 850 K.

Response: Thanks for your suggestion. We also tried higher temperature (above 823K), under which, the samples show signs of the Se volatilization after the measurement, we observe the thermocouple of ZEM-3 turned black indicating contamination. Thus, to protect our facility, all property measurements were executed below 800K.

The following sentence has been added in the revision, on p.9, l. 53-57 under the “**Electrical and thermal transport property measurements**” section: “**Note that all the property measurements were executed below 800 K to protect the facilities since at the higher temperature the samples may undergo Se volatilization which will contaminate the thermocouple probe.**”

2. How about the stability related to the current density? The stability depends on the applied electronic field should be provided, especially the critical voltage.

Response: Thanks. Faithfully following your suggestions, we tested the current density (J) dependence of the relative electrical resistance variation (R/R_0) for both the hot-pressed-only- and the hot-pressed-annealed- Ag_9GaSe_6 , $\text{Ag}_{8.3}\text{Cu}_{0.7}\text{GaSe}_6$ and $\text{Ag}_9\text{GaSe}_{5.5}\text{Te}_{0.5}$ at 750K. A new section “critical voltage determination” was added in the Supporting Information. As new Fig. S10 reveals, the R/R_0 declines almost linearly as J increases for the hot-pressed-only-samples, but shows firstly a plateau and then decreases sharply for the hot-pressed-annealed Ag_9GaSe_6 and $\text{Ag}_9\text{GaSe}_{5.5}\text{Te}_{0.5}$ samples indicating the existence of a critical voltage (V_c). (V_c is first defined in *Nat. Commun.* **2018**, *9*, 2910) The appearance of V_c after the annealing demonstrates a stability enhancement. However, even after the annealing treatment, the Cu-doped $\text{Ag}_{8.3}\text{Cu}_{0.7}\text{GaSe}_6$ sample doesn't show a V_c (Fig. S10), this is probably because that 1): $\text{Ag}_{8.3}\text{Cu}_{0.7}\text{GaSe}_6$ undergoes a phase transition at 330K as reported in Ref. 27 (Fig. S2); and 2): Cu atom is smaller than Ag, which is more mobile that aggravates the sample instability, so that the simple annealing treatment is not as helpful and effective as those observed in samples of Ag_9GaSe_6 and $\text{Ag}_9\text{GaSe}_{5.5}\text{Te}_{0.5}$. Nevertheless, the reproducibility of TE properties of the Cu-doped sample is improved as shown in the main text.

Moreover, in Fig. S11, the V_c of the annealed Ag_9GaSe_6 (0.046V) and $\text{Ag}_9\text{GaSe}_{5.5}\text{Te}_{0.5}$ (0.072V) lie between those of Cu_2S (~0.02V) and Cu_2Se (~0.11V), indicating the stability of our sample is comparable to that of the Cu_2Q -based liquid-like TE materials (*Nat. Commun.* **2018**, *9*, 2910).

The following sentences are added in the revision on p. 5, l. 30: “**Moreover, the critical voltage (V_c) was measured to be 0.05 and 0.07 V on the hot-pressed-annealed- Ag_9GaSe_6 and the hot-pressed-annealed- $\text{Ag}_9\text{GaSe}_{5.5}\text{Te}_{0.5}$, respectively, (Fig. S10) which lies between those of the state-of-the-art liquid-like TE materials, Cu_2S (0.02 V) and Cu_2Se (0.11 V),⁴² indicating that the subsequent annealing treatment enhances the sample stability. (Fig. S11)”**

Fig. S10. Current density dependence of relative electrical resistance variation (R/R_0) for the hot-pressed-only- and hot-pressed-annealed- Ag_9GaSe_6 , $\text{Ag}_{8.3}\text{Cu}_{0.7}\text{GaSe}_6$ and $\text{Ag}_9\text{GaSe}_{5.5}\text{Te}_{0.5}$. The vertical dotted line indicates the ending of the R/R_0 plateau and starting of the sharp decrease of the R/R_0 curve, indicating the existence of a critical voltage (V_c).

Fig. S11. Experimentally determined V_c of Ag_9GaSe_6 and $\text{Ag}_9\text{GaSe}_{5.5}\text{Te}_{0.5}$ at 750 K and the reported values of Cu_2S and Cu_2Se^3 are also presented.

3. The theoretical calculation should be carried out to clearly show the migration paths of Ag ions.

Response: Thanks for your suggestion. We have tried the corresponding calculations, but run into the following difficulties that hinders the theoretical simulation at present:

1) The Ag atoms are highly disordered in the cubic Ag_9GaSe_6 , the complicated occupancies of Ag atoms lead to a great difficulty in modeling. To date, the calculations reported in literature, merely consider the case where the Ag atoms are fully ordered in the low-temperature phase (space group $P2_13$). (*Joule* **2017**, *1*, 816; *Mater. Today Phys.* **2018**, *5*, 20)

2) In terms of the calculation method, the ab initio static calculation can only simulate the influence of the pressure, but not the temperature. However, our actual experimental conditions are under the high temperature and simultaneously under the high pressure.

Therefore, to accurately calculate and simulate the experimental conditions, needs extra efforts, which will be done with cooperation with other experts in the future.

4. TEM are strongly recommended to observe the possible precipitation of Ag nanoparticle.

Response: Faithfully following your suggestions, SEM observations are supplied. A new Fig. S7 is provided in the Supporting Information. As shown, the hot-pressed-annealed-sample shows homogenous and clean cross section, and a nearly constant Ag content 50.63% vs 51.19%, after the measurement; on the contrary, without the annealing treatment, the hot-pressed-only sample show heterogeneous particles, an uneven surface, and a considerable Ag element enrichment from 52 to 63% after the TE-property measurement. Similar Ag-enrichment is observed (from 52–55% to 64–66%) by Luo et al. (*Chem. Eng. J.* **2019**, 374, 494). Therefore, the positive effect of annealing to stabilize the Ag_9GaSe_6 is reconfirmed.

As you suggested, the following sentences are added in the main text on p. 4, l. 31-35: “... Besides, the scanning electron microscopy observation reveals that the Ag-rich precipitation is observed on the hot-pressed-only- Ag_9GaSe_6 after the TE-property measurement (Figs. S7a vs b), manifesting the sample instability as similar as that observed by Luo.²⁷...”

and on p.4, l.57-61: “... Moreover, the SEM observation reveals no obvious Ag precipitation in the hot-pressed-annealed sample before and after the TE property measurements, indicating the phase and composition stability of the samples. (Fig. S7c vs d)”

New Fig. S7. a–d, SEM images of the cross section of the hot-pressed-only and hot-pressed-annealed Ag_9GaSe_6 samples before and after the 4th TE-performance measurement run, respectively. e–h, the corresponding elemental distribution scanning spectra of a–d with Ag content listed.

Reviewer #1 (Remarks to the Author):

The authors have properly addressed all my concerns. I think it is suitable for publication now.

Reviewer #2 (Remarks to the Author):

The responses the reviewer comments about the manuscript entitled "Ag₉GaSe₆: High-pressure-induced Ag migration causes thermoelectric performance irreproducibility and elimination of such instability" is not well explained and not sufficient to publish it in the Nature Communications. For example, in the Fig. S9 elemental distribution mapping indicates doubtful data like before and after annealing authors got exactly similar bright spots (red circled) in the elemental distribution mapping although SEM images are different. How is this even possible when authors are performing SEM of different places of sample ?

Fig. S9

Furthermore, exactly similar, and even higher thermoelectric performance in the same material is already reported in the literatures (*Chem. Commun.* **2017**, 53, 11658–11661 and *Chem. Eng.*

J. 2019, 374, 494–501). Jiang, B. *et al.* already mentioned about the liquid-like migration of Ag ions in Ag_9GaSe_6 (*Chem. Commun.* **2017**, 53, 11658–11661) with similar ZT ~ 1.3 at 800 K.

K. Qi, X. et al. showed the same percentage Cu-doping (*Chem. Eng. J.* **2019**, 374, 494–501) in Ag_9GaSe_6 ($\text{Ag}_{8.28}\text{Cu}_{0.72}\text{GaSe}_6$) and obtained a ZT of ~ 1.6 at 824 K and ultralow thermal conductivity. Eventually Qi, X. *et al.* also did the annealing treatment as reported in the present manuscript and mentioned “The annealing treatment can be utilized to recover the composition change and make the sample homogeneous again.” Therefore, the work presented in this manuscript does not show any improvement in thermoelectric figure of merit in comparison to the earlier reported literature as well as no new idea is presented. In terms of novelty it is not suitable for the publication in Nature Communications.

Reviewer #3 (Remarks to the Author): The authors tried their best to respond the comments and revise the manuscript accordingly. Thus, I suggest acceptance.

R1:

The authors have properly addressed all my concerns. I think it is suitable for publication now.

Response: Thank you very much. We appreciate the positive comments and valuable suggestions.

R2:

Q1: The responses the reviewer comments about the manuscript entitled "Ag₉GaSe₆: High-pressure-induced Ag migration causes thermoelectric performance irreproducibility and elimination of such instability" is not well explained and not sufficient to publish it in the Nature Communications. For example, in the Fig. S9 elemental distribution mapping indicates doubtful data like before and after annealing authors got exactly similar bright spots (red circled) in the elemental distribution mapping although SEM images are different. How is this even possible when authors are performing SEM of different places of sample?

Previous Fig. S9

Response: These so-called “similar bright spots” are the default settings of the element label of the instrument. We reported as received and had not edited, please see enlarged images in the following Coverlet Fig. 1. Thank you for raising this issue. To avoid such an unnecessary misunderstanding, in the revision, a new Fig. S9 is provided in which the element label format is improved.

CoverlettFig. 1 Enlarged image of the elemental distribution mapping section in the previous Fig. S9. The element labels are shown as accepted by the instrument default setting.

New Fig. S9 in which the element label format is adjusted to avoid an unnecessary misunderstanding.

Q2: Furthermore, exactly similar, and even higher thermoelectric performance in the same material is already reported in the literatures (*Chem. Commun.* **2017**, 53, 11658–11661 and *Chem. Eng. J.* **2019**, 374, 494–501). Jiang, B. *et al.* already mentioned about the liquid-like migration of Ag ions in Ag_9GaSe_6 (*Chem. Commun.* **2017**, 53, 11658–11661) with similar $ZT \sim 1.3$ at 800 K. Qi, X. *et al.* showed the same percentage Cu-doping (*Chem. Eng. J.* **2019**, 374, 494–501) in Ag_9GaSe_6 ($\text{Ag}_{8.28}\text{Cu}_{0.72}\text{GaSe}_6$) and obtained a ZT of ~ 1.6 at 824 K and ultralow thermal conductivity.

Response: Firstly, our aim is to study the intrinsic mechanism of the stability of Ag_9GaSe_6 , which has never been deeply understood before. Therefore, we purposely chose the best reported Ag_9GaSe_6 -based materials, which are mentioned above, as cited in our manuscript ref. 15, 27, respectively, to prove that within the **identical materials**, we not only can realize the high ZT values as reported, but also the reproducibility and stability, which has **never been reported**.

For the liquid-like thermoelectric materials, the common problem is the most unwanted poor-stability and thermoelectric performance irreproducibility, which were reported to be evidenced by metal particles that are as-decomposed and deposited on the sample pellet even after the 1st measurement run.

In this manuscript, we demonstrate such a poor-stability and irreproducibility are caused by the pressure-driven Ag migration to the crystallographic sites with higher-chemical potentials. Using the previously reported pristine- and doped- Ag_9GaSe_6 based-materials as examples, we show such instability can be eliminated by a simple annealing treatment.

Q3: Eventually Qi, X. *et al.* also did the annealing treatment as reported in the present manuscript and mentioned “The annealing treatment can be utilized to recover the composition change and make the sample homogeneous again.” Therefore, the work presented in this manuscript does not show any improvement in thermoelectric figure of merit in comparison to the earlier reported literature as well as no new idea is presented. In terms of novelty it is not suitable for the publication in *Nature Communications*.

Response: We provided 43 sets of single crystal diffraction data collected on the as-obtained Ag_9GaSe_6 samples to support our discussion and conclusion. Single crystal diffraction is a useful and widely applied technique commonly utilized in MOF, coordination chemistry, NLO material, or battery material fields, etc., however, in the thermoelectric community, which has been less used.

In the mentioned literature, (Qi, X. *et al.*, *Chem. Eng. J.* **2019**, 374, 494–501, also the ref. 27 in our manuscript) indeed reported the electrical properties of their “annealed sample”. As shown in Coverlet Fig. 2, their annealed sample still exhibits poor reproducibility indicated by the large deviations between the 1st and 2nd run of tests of the electrical conductivity and Seebeck coefficient, respectively.

Moreover, Qi, X. *et al.* in ref. 27 only report that the annealing treatment makes the sample return to the original state, but doesn’t address why the unrepeatability and instability is still observed, as they have written “It is found that the annealed sample is very close to the as-prepared one because of the similar the electrical conductivity and Seebeck coefficients”, which indicates in their sample

no stability improvement is realized after annealing. In sharp contrast, our data show reproducibility and stability after annealing. (CoverlettFig. 2) Because the annealing procedure is different. In our work, the annealing procedure is holding at 823 K for 24 h under N₂ flow in the furnace; while in the literature ref. 27, “...the sample Ag₉GaSe₆ was isothermally annealed at 823 K for 30 mins in the graphite crucible assembled in a hot press instrument after ZEM-3 measurement”, the annealing is under pressure.

Our discovery demonstrates that the pressure during the annealing reported in ref. 27. cause the unwanted instability.

CoverlettFig. 2 The electrical conductivity and Seebeck coefficient of the annealed Ag₉GaSe₆ sample after several runs of measurement: **a, b**, ref. 27 (*Chem. Eng. J.* **2019**, 374, 494–501). **c, d**, this work.

R3:

The authors tried their best to respond the comments and revise the manuscript accordingly. Thus, I suggest acceptance.

Response: Many thanks.